# Animal Brucellosis in Egypt: Review on Evolution, Epidemiological Situation, Prevalent *Brucella* Strains, Genetic Diversity, and Assessment of Implemented National Control Measures

**DOI:** 10.3390/microorganisms13010170

**Published:** 2025-01-15

**Authors:** Ahmed M. S. Menshawy, Acacia Ferreira Vicente, Yamen M. Hegazy, Vitomir Djokic, Mahmoud E. R. Hamdy, Luca Freddi, Essam M. Elbauomy, Ashraf E. Sayour, Claire Ponsart, Nour H. Abdel-Hamid

**Affiliations:** 1Department of Animal Medicine, Faculty of Veterinary Medicine, Beni-Suef University, Beni-Suef 62511, Egypt; elmenshawy81@gmail.com; 2WOAH/EU & National Reference Laboratory for Animal Brucellosis, ANSES/Paris-Est University, 94700 Maisons-Alfort, France; acacia.ferreiravicente@anses.fr (A.F.V.); vitomir.djokic@anses.fr (V.D.); luca.freddi@anses.fr (L.F.); claire.ponsart@anses.fr (C.P.); 3Department of Animal Medicine, Faculty of Veterinary Medicine, Kafr El-Sheikh University, Kafrelsheikh 33516, Egypt; yamen12@yahoo.com; 4Department of Clinical Studies, College of Veterinary Medicine, King Faisal University, Al-Ahsa 31982, Saudi Arabia; 5WOAH Reference Laboratory for Brucellosis, Department of Brucellosis Research, Agricultural Research Center, Animal Health Research Institute, P.O. Box 264-Giza, Cairo 12618, Egypt; merhamdy@hotmail.com (M.E.R.H.); elbauomyessam@yahoo.com (E.M.E.); hossam.sayour@gmail.com (A.E.S.)

**Keywords:** brucellosis, control, Egypt, livestock, MLVA16, prevalence, WGS

## Abstract

Brucellosis is a neglected zoonotic disease that has a significant economic and public health impact, especially in endemic countries. This review delves deeply into brucellosis’s current epidemiological situation and potential sources of livestock infection in Egypt during the last two decades. MLVA-16 and Whole Genome Sequencing based on core-genome SNP analyses confirm the presence of different *B. abortus* and *B. melitensis* outbreak strains, both older widely disseminated *Brucella* strains and newly introduced ones. Despite implementing the test-and-slaughter control strategy over forty years, the disease is still endemic, and different *Brucella* species circulate among several animal species. The raising of mixed animal species in the same households or farms, exposure to aborted animals, and lack of public awareness about brucellosis transmission are among the main risk factors for increasing livestock brucellosis prevalence in Egypt. Young animals’ voluntary vaccination, lack of a nationwide animal identification system, and uncontrolled animal movement stand beyond the ineffectively applied control strategy and may be subdued by applying mass vaccination to decrease disease prevalence dramatically and target imported camels, domestic pigs, and dogs (housed and stray) in the national control surveillance. Increasing awareness through educational campaigns is compulsory to reduce brucellosis transmission risk to livestock/humans.

## 1. Introduction

Brucellosis, caused by members of the *Brucella* genus, is a major re-emerging disease that affects livestock, domestic animals, wildlife, and marine mammals [1]. It causes re-productive disorders, including late-stage abortions, stillbirths, low milk production, in-fertility, and placental retention in females. In addition, orchitis and epididymitis in males [2].

Brucellosis is considered one of the most widespread zoonoses worldwide [3]. Symptoms of human infection are mostly non-specific. Some complications may emerge mainly due to antibiotic-resistant strains or antibiotic monotherapy. It is not easy to establish the exact incidence of human brucellosis worldwide. However, the annual global incidence is estimated to be 2.1 million, with Africa and Asia sustaining most of the world-wide risk and cases [4]. The genus *Brucella* currently comprises 12 species [1,5,6]. Four *Brucella* species, *B. melitensis*, *B. abortus*, *B. suis*, and *B. canis*, were identified in Egypt [7,8,9,10]. However, brucellae are host-adapted to preferred hosts. They can infect other species due to cross-species transmission [11,12]. For instance, *B. abortus*, which is host-adapted to cattle, can infect small ruminants [13,14] and wildlife [15], complicating the epidemiology of the disease and its control measures.

The disease has a worldwide dissemination. However, some countries, especially high-income nations, are considered free from both *B. abortus* and *B. melitensis*. Brucellosis is often underreported in developing countries, with inadequate or insufficient management measures, resulting in considerable health, economic, and livelihood burdens [1,3].

In Egypt, Brucellosis was first reported in 1939. The disease was established then and has since been detected in humans and different animal species, especially ruminants, causing great endemicity until now [16]. In the late 1970s and early 1980s, the government initiated an open-door policy to compensate for the shortage of animal proteins accompanied by the importation of dairy cattle from many different countries without testing animals before importation at the time. During this period, brucellosis had also been reported among buffaloes, sheep, goats, horses, donkeys, swine, dogs, rats, and camels [17]. Between 1985 and 1987, the alarming figures of brucellosis incidence and prevalence in many dairy farms were the main reason for the bilateral American-Egyptian project (EG-APHIS-217). Across this project, the control program consisted of a test-and-slaughter policy in the targeted farms, vaccination of calves using the S19 vaccine, vaccination of adults using K45/20, and quarantine for the newly purchased animals for 30 days. On the other hand, sheep and goats received little attention in this national control program in the same period, and this is expected to be the reason behind the wide distribution of *B. melitensis* among different ruminant species in the country [17].

At the same time, the general veterinary authorities in Egypt have established a control program for the eradication of brucellosis depending mainly on the test and slaughter of adult seropositive and young animals vaccinated with the S19 vaccine, after the application of a battery of serological techniques for detection of positive animals according to the procedures officially validated by World Organization for animal health (WOAH) concurrently with S19 vaccination. However, the use of such a vaccine hampers serodiagnosis because the vaccinated and naturally infected animals are difficult to distinguish [18].

Currently, vaccination against bovine brucellosis in Egypt is noncompulsory according to ministerial decree 102/2017, where 4–7 months female cattle and buffaloes are vaccinated using *B. abortus* strain 19 while 4–7 months female sheep and goats are vaccinated using *B. melitensis* strain Rev.1. Side by side, the rough *B. abortus* strain RB-51 vaccine is used by some private farms under direct supervision of a veterinary authority [19]. However, the protection given by the rough vaccine RB51 was insufficient, and field *B. abortus* bv1 and *B. melitensis* bv3 have been identified bacteriologically and molecularly from aborted fetuses of aborted cattle belonging to farms vaccinated with RB51 [20].

The prevalence of brucellosis in Egypt ranges widely from 0.2% to 20% among animals, depending on the official and different survey research studies based on the study area or the animal species [21,22,23,24,25].

Control strategies for bovine brucellosis were implemented in Egypt since the 1980s. Despite the high public health and economic importance of brucellosis, the national Egyptian brucellosis control program does not appear to have the full capability for the complete eradication of the disease [10,12,16,26]. As a result, there has been a marked increase in *B. melitensis* prevalence in atypical hosts like cattle, buffaloes, and camels due to trans-species transmission of *B. melitensis*, especially under the mixed husbandry system, which is the most frequent in Egypt [12,27,28,29,30].

In Egypt, human brucellosis incidence data are scarce, not easy to gather, and are unreliable because of misdiagnosis; cases are frequently reported as pyrexia of unknown cause, and not all patients seek treatment at public hospitals. Therefore, the disease is probably underreported [23].

In previous surveillance of brucellosis in humans in Egypt, the annual incidence of infected cases ranged between 0.5/100,000 and 70/100,000, depending on the region of study [29,31,32]. This high incidence might be attributed to the low biosecurity inside the farms, high livestock and human density, predominance of smallholdings that favor close contact between humans and animals, exposure to abortion, presence of mixed populations of different species at the same place, as well as some faulty traditional habits such as consumption of raw milk and unpasteurized dairy products, which represent the major risk factors for persistence of brucellosis in Egypt [23,30,33,34,35]. The complicated epidemiological situation of brucellosis in Egypt could be reflected through the isolation of bacteria from atypical hosts such as *B. melitensis* from large ruminants [28,36] and from Nile catfish [37] and *B. abortus* from dogs and cats [38]. Also, the human pathogenic *B. suis* bv1 was recovered from cows’ milk [28]. Furthermore, *B. canis*, *B. suis* bv2, and *B. melitensis* were identified as neglected animal species [8,9,39] that were not involved in the national surveillance systems like dogs, pigs, and camels, which necessitates immediate and regular surveillance by GOVS authorities for early detection and control of *B. suis* and *B. canis* biovars spread in Egypt.

## 2. Seroprevalence of Brucellosis in Egypt

We separated the present review into two periods (2000–2010) and (2011–2024) to make it convenient and easier to analyze and discuss the enormous amounts of data at each time. In addition, the MALVA analysis and whole-core-genome single-nucleotide polymorphism (cgSNP) analysis of recovered *Brucella* strains in Egypt occurred after 2010.

### 2.1. Brucellosis Frequency and Control Between 2000 and 2010

The national control measures were revised in 1999, and the elements that have been changed included sheep and goats in the testing regimens as well as slaughtering of the seropositive animals with compensation to the owners. Moreover, lambs and goat kids vaccinated with the Rev 1 vaccine were incorporated into the control measures, but the vaccination for all species was voluntary. Finally, the GOVS stopped vaccinating adults with K45/20 [17], as the expressed immunity level was low, and animal vaccination re-quires booster doses and repeated annual dosing. On the other hand, young ruminants were vaccinated with smooth *Brucella* vaccines (S19 or Rev1) that provoked life-long im-munity. During this period, there is an obvious discrepancy between official seroprevalence data and data from scientific publications.

#### 2.1.1. Governmental Estimates

The official reports showed that brucellosis is still endemic among large and small ruminants in Egypt. The General Organization for Veterinary Services (GOVS) is implementing a surveillance program for Brucellosis based on passive surveillance and risk-based surveillance. The passive surveillance system is based on the requirement that all stakeholders immediately report any indication of disease to the nearest veterinary clinics. Veterinary clinics must notify the local veterinary authority, then the central veterinary authority, the General Organization for Veterinary Services (GOVS). To promote and support timely notice and response to any disease suspicions, the transboundary animal disease information system in epidemiological units receives all epidemiological data from veterinary clinic records. GOVS set up a hotline to receive any notifications. GOVS receives notice warnings by email or fax, as well as monthly reports sent from all governorates revealing the health situation in terms of zoonotic illnesses. Passive surveillance reports are submitted to WOAH-WAHIS every six months.

Controlling the disease and reducing the chance of animal-to-human transmission are the primary goals of risk-based surveillance for brucellosis. This nationwide campaign, which relies on serological testing and culling of positive reactor animals, is governed by Ministerial Decree 1329/1999. The total number of animals that should be tested annually is estimated to be 10% of the total animal populations at high risk. The surveil-lance is targeted and implemented in both farms and backyard sectors of at-risk geo-graphical areas with a high prevalence of brucellosis based on previous results of different types of surveillance. The risky geographical districts have high-density animal populations, particularly the mixed backyard population. Based on prior surveillance results, the surveillance is targeted and implemented in farms and backyards of at-risk geographical areas with a high prevalence of brucellosis in geographical localities at high risk with dense animal populations, particularly, mixed backyard populations.

Data on the active surveillance that has been carried out by the GOVS during the period from 1999 to 2011 were collected by Eltholth et al. [40]. A total number of 4,331,154 animals were tested between 1999 and 2011. The estimated prevalence did not exceed 1.32% in any year among different ruminant populations in Egypt. The proportion of examined cattle, buffaloes, sheep, and goats did not exceed differences of 4.5%, 2%, 2%, and 1% of the target population in any year of the observation period. Sheep and goats recorded higher prevalence every year compared to cattle and buffaloes [40]. There are many studies for the surveillance reports obtained by the national control program during this period and all of them use the same figures obtained by Eltholth et al. [40]; Hegazy et al. [33] in Upper Egypt; Eltholth et al. [41] in Behira.

#### 2.1.2. Research Estimates

The summary of figures obtained in various studies during this observation period was explored in Figure 1. Between 2000 and 2010, the prevalence of brucellosis in cattle and buffaloes was significantly higher in different Egyptian regions, ranging from 0.79% to 15% in some governorates representing Upper Egypt, the Nile Delta region, and Egyptian boundaries [26,27,42,43,44,45]. The corresponding estimates of brucellosis seroprevalence in sheep and goats were much higher than in cattle and buffaloes, where it ranged from 2 to 20% in sheep [26,27,44] and 3 to 21% in goats [27,42,43,44,46].

The estimated seroprevalence of brucellosis in camels was 7% to 9% [47,48]. This high prevalence estimate was caused by some seropositive camels evading the quarantine and a lack of governmental control measures for camels.

In this period between 2000 and 2010, the governmental underestimated the brucellosis situation in the country, as it is not clear what the criteria for the selection of the herds and animals tested in this survey were [13,33]. The lack of random sampling and the systematic approach in selecting the animals for testing against brucellosis hindered the accurate calculation of brucellosis prevalence both in herds and flocks and in animals. The rate of positive per tested animals, according to governmental reports over 2008, varies from 0% to 2.44%, which is an under-estimation, as they claimed in the official reports compared to scientific studies [33,49]. Various field surveys implemented in Egypt indicate a much higher prevalence. On the other hand, some of the re-search sample size estimates had issues of biases in random sampling designs. Some studies found that the testing regimens were not formally accepted by GOVS or WOAH. As a result, these estimations must be regarded with extreme caution, as the samples appear to have been conveniently or purposefully picked. There was no evidence that the sampling was implemented using systematic or probabilistic methods.

### 2.2. Brucellosis Frequency and Control Between 2011 and 2023

#### 2.2.1. Governmental Estimates

The lack of sufficient governmental information on brucellosis prevalence during this period may be attributed to the prior outbreak events of Avian Influenza in 2006, swine flu in 2008, many foot-and-mouth disease (FMD), lumpy skin disease (LSD), and bovine ephemeral fever (BEF) outbreaks in animals, and the human pandemic COVID-19 in 2019. These outbreaks have drawn the responsiveness of the GOVS which paid little attention to the national control program of brucellosis. This was also reported by WOAH in 2020, which showed that Egypt had not recorded bovine brucellosis in that year, but the bovine brucellosis is still endemic at low prevalence estimates WOAH (WOAH website). It is clear that the official numbers of brucellosis cases among various animal species in Egypt reported to WOAH (OIE website) every 6 months from 2005 to 2019 is almost similar and fixed; almost 500 new cases of ruminant brucellosis every 6 months.

#### 2.2.2. Research Estimates

The research studies carried out in this observation period to estimate brucellosis prevalence among different ruminant species have increased dramatically. These estimates are plotted in Figure 2 and Figure 3. The estimates presented in these figures do not include estimates of the studies carried out in slaughterhouses, and if there is more than one study carried out in the same governorate, the lowest prevalence was shown in the figures.

#### Cattle and Buffaloes

The prevalence of brucellosis among cattle is reported in (Figure 2) to be high in different governorates of Egypt. Most of the researchers reported prevalence estimates of 5% to 10% in Nile Delta Governorates [50,51,52,53,54,55,56,57,58,59], in Upper Egypt governorates [60], and Egyptian boundary governorates [61,62]. Moreover, some studies reported brucellosis prevalence higher than 10% in North Sinai [63] and the Nile Delta region [64], and in upper Egypt [65].

#### Sheep and Goats

The individual prevalence estimates in small ruminants differed from those in cattle and buffaloes. Most of the studies (Figure 2) reported a prevalence higher than 10%; in Kafrelsheikh, Gharbia, Dakahlia, and KafrElsheikh, Nile Delta governorates [23,24,33,51,54,64]; in Giza, Abdel-Hamid et al. [21], in Matrouh and North Sinai, Egyptian boundary governorates [62,66,67], and in Fayoum, Assuit, Almynia, Upper Egypt governorates [32,68]. Studies that reported a prevalence lower than 10% include Ghanem [69] at Kafrelsheikh; Al-Habaty et al. [70] at slaughterhouses in Assiut governorate; Hashem et al. [55] in the Nile Delta (Damietta and Dakahlia).

#### Camels

Most of the camels in Egypt nowadays are imported for slaughtering. Therefore, there is a marked discrepancy between different reports on brucellosis prevalence in camels. Camels are not involved in the national surveillance programs in Egypt. Thus, there are no official reports about the brucellosis seroprevalence among the native camels. The prevalence estimates of the published research ranged from 3.5 to 20% in the Egyptian boundary governorates of North Sinai, Matrouh, New Valley, and Red Sea governorates [62,67,71,72,73,74], governorates of the Nile Delta [75,76,77,78], and governorates of Upper Egypt [64,71,76,79,80] (Figure 3).

Some of these results were based on a sound epidemiological base of random sampling, and these estimates offered a clear picture of disease prevalence among animals and governorates included in these articles. These studies include El Sherbini et al. [42], Hegazy et al. [33], Holt et al. [34], Abdel-Hamid et al. [21], Hegazy et al. [24], Ramadan et al. [58], and Khalafalla et al. [53]. Other articles depending on the outbreak investigations or convenience sampling cannot offer very accurate estimates of individual dis-ease prevalence. Moreover, these studies could not consider the sensitivity and specificity of the serological tests in the estimation of prevalence, and this may under-estimate the actual situation of brucellosis prevalence. On the other hand, these articles could show that the magnitude of disease spatial spread in the country without the herd prevalence obtained in these studies is very high and indicate the wide spread of the disease in the country.

The current brucellosis infection in Egypt exemplifies the unsuccessfulness of the efforts to control a disease for which effective tools are readily available, as shown in a proposed Figure 4 below. Between the 1980s and 1990s, there was no obvious discrepancy between the official seroprevalence reported by GOVS data and the data offered through scientific publications (Figure 4). The discrepancy between the official seroprevalence estimates and those reported by the published research became obvious and significant over time. The veterinary authorities may not have used clear criteria to select the herds and animals tested in the surveys, resulting in no random sampling and a planned selection of animals for brucellosis testing. This makes it harder to obtain an accurate picture of brucellosis prevalence in herds, flocks, and individual animals. Also, sampling bias in much published scientific research could be another reason. Establishing goals for the implementation of animal disease control strategies that are adapted to the level of available resources, suitable for the structure of the animal production system, and sensitive to the incentives for disease control of the different stakeholders are prerequisite elements for a successful national control program.

## 3. Causes of National Control Program Unsuccessfulness to Control Brucellosis in Egypt

Effective control of brucellosis requires some essential elements such as regular estimation of the disease frequency to identify infected animals and herds, prevention of transmission to non-infected animals and herds, and eradication of the reservoirs to eliminate the sources of infection for susceptible animals or to prevent re-introduction of the disease to a free herd [10,12,81,82]. None of these elements for brucellosis control are being applied in Egypt. First, as shown in the previous section of this review, the surveillance system in Egypt is not able to identify the actual situation of the disease.

### 3.1. Inappropriateness of the Planned Control Measures for Brucellosis Control in Egypt

The Food and Agriculture Organization has set guidelines for brucellosis control and eradication of animal brucellosis in different localities based on their prevalence estimates. These guidelines for brucellosis control in a region must depend on many factors, such as disease prevalence in the herds or flocks, type of animal husbandry, economic resources available for control, public health influences, and the possible international trade consequences. Policymakers’ decisions are likely to be intuitive unless accurate and current epidemiological information is available. As shown in the previous section of this review, the epidemiological information available in Egypt proves widespread disease in the whole country with elevated prevalence estimates in some localities either on the individual or the flock level. Therefore, the choice of GOVS in Egypt for applying a test-and-slaughter policy in parallel with voluntary vaccination of young ruminants, which should be undertaken in areas with animal brucellosis prevalence < 2% for disease eradication, is not considered to be the appropriate strategy for disease control in Egypt [10,33,40].

The national control program elements are to test all female ruminants older than 6 months of age yearly against brucellosis and slaughter adult seropositive animals with voluntary vaccination of young calves, lambs, and kids. The government laboratory in Abbassyia produces far fewer vaccines than are required. The annual production rate of the *B. abortus* S19 vaccine for cattle and buffaloes is approximately 50,000 doses, while the Rev1 vaccine for sheep and goats is approximately 5000 doses [49].

Additionally, the official reports, as shown before, and research studies have shown that the GOVS was not able to test more than 5% at the maximum of the proposed number of all ruminant species annually [26]. The sampling is being carried out in various percentages among different ruminant species and districts. In almost all districts, cows and buffaloes were regularly sampled while sheep and goats were randomly sampled. These findings may result from the unrealistically high sampling targets, given the resources available, poor farmer compliance with the official control measures, and the structure of the Egyptian production system [25,26,83]. Lack of funding, as reported by Refai [17], Eltholth et al. [40], and Eldeihy et al. [83], often prevents the continuum of these control campaigns, and this may be the reason for not reaching the official sampling targets. The shortage of veterinary clinicians and lack of incentives could also be a probable cause for not adhering to the national control measures. Poor compliance from the owners is a major cause of such findings, as they refuse to slaughter their positive animals because of the low compensation value and the payback delays [83]. All of these obstacles led to the low level of testing and the infrequent sampling protocols that occurred in such control campaigns which failed the control program at the current level to minimize the brucellosis prevalence among ruminants even with the continuous campaign for the last 20 years in a simulated scenario as shown in findings by [26].

The principal causes for the applied control strategy’s unsuccessfulness in Egypt may be due to the insufficient coordination, cooperation, and communication between the veterinary and the public health authorities, the application of voluntary vaccination that did not cover all young animals representing 15–25% of the total animal population, inappropriate animal identification system particularly in some animal species, and lack of the control on animal movement.. Supporting this scenario are the two peaks of brucellosis cases among ruminants representing two outbreaks in 2008 and 2016 (Figure 5) in Egypt, represented by a time series graph based on data downloaded from WAHIS (https://wahis.woah.org/#/dashboards/qd-dashboard, accessed on 20 November 2024) and analyzed by StataCorp version 18 [84]. The official reports detail six-monthly immediate notifications and follow-up reports over 19 years of passive surveillance (2005–2023). These findings may be attributed to vaccine under coverage (voluntary vaccination) where the General Organization of Veterinary Services (GOVS) vaccinated approximately 50,000 cattle and 50,000 sheep only by S19 and Riv1 vaccines every year during this period [40,85].

Many other scenarios could be applicable and result in controlling brucellosis among ruminant species in Egypt, as indicated by Hegazy et al. [12]. These scenarios could be of low cost and minimum usage of human resources and may be acceptable to the farmers. The study results showed that high vaccination coverage for young replacement animals could significantly reduce the prevalence of small ruminant brucellosis in endemic areas. On the contrary, the test-and-slaughter strategy alone is not able to control small ruminant brucellosis in these endemic settings with low-income resources. Moreover, other strategies requiring a relatively low overall vaccination coverage, such as the vaccination of 50% of young replacements and 25% of adult animals, could represent potential successful options in these areas.

The second aspect of brucellosis control, which is not adhered to by GOVS and contributes to the dissemination of the infection, is the prevention of transmission to non-infected animals.

### 3.2. Risk Practices by the Farmers (Shepherds) and Also Veterinarians

The survey carried out to obtain information on the knowledge, attitudes, and practices of the shepherds and farmers on brucellosis showed that there is a lack of compliance with the national control program [24,83]. Results of such surveys showed that high-risk practices are applied among shepherds and farmers, as well as a lack of sufficient information to prevent and control the disease. A total of 71.4% of farmers keep animals that had miscarriages, and 23.8% would sell them in markets. Also, 55.5% fed aborted materials to dogs, 27.8% threw them in water canals, and 16.7% buried them [24].

All of the farmers in that survey [83] confirmed that there was no sampling from the GOVS or vaccination against brucellosis in their villages at all. The way of animal husbandry in Egypt also is considered as a great risk of disease spread and failure of the control program like the existence of production systems that allow regular mixing of small ruminants from different households/flocks [86]. Also, the inability to regulate animal movement and lack of animal identification are likely to be limiting factors for the effectiveness of a brucellosis control program [10]. This was confirmed through different published research where the same *Brucella* genotype where isolated from different Egyptian governorates as well as different animal species [7,10,87]. This free movement pattern, close contact between animals of different species and origins, close contact between humans and infected/aborted animals, and consumption of unprocessed dairy products are key determinants for the transmission of contagious diseases within the animal population and zoonotic diseases such as brucellosis between animals and humans [35,88]. Most sheep or goat flocks in Egypt are mobile. Movement of infected sheep or goats can contaminate pastures and spread brucellosis to other animals. Hegazy et al. [12] and Etman et al. [89] confirm that uncontrolled animal trade/movement between villages significantly impacts the efficacy of any brucellosis control program. Control of animal movements is fundamental and mandatory to the applied combined control strategy, next to proper animal identification [10].

Moreover, the level of test and slaughter in Egypt, which does not include more than 5% of the target population every year, leaves the chance for infected animals to come in contact again with areas from which the disease has been eliminated and spread the infection again.

### 3.3. The Genetic Diversity and Strain Relatedness of B. melitensis in Egypt Under Various Husbandry Systems in Different Animal Species

*B. melitensis* bv3 was the predominant strain isolated from the typical (small ruminants) and atypical hosts (large ruminants) [12,22,90,91]. Recently, Sayour et al. [7] and Hegazy et al. [12] concluded that *B. melitensis* could circulate among the cattle population via cow-to-cow transmission in the absence of sheep and goats (preferential host). This means that targeting one species by controlling either sheep or cattle is going to have a low effect on disease control. Furthermore, more than one genotype from *B. melitensis* bv3 isolates could be obtained from the same animal, and the cause of that may be a combination of different animal species in the same household or farms and the lack of cross-protection between genotypes which puts into question the efficacy of vaccines against brucellosis.

### 3.4. Biological Causes

Finally, many biological causes are considered limiting factors for test-and-slaughter policies to succeed in different countries, especially in developing countries with animal husbandry practices such as Egypt. A total of 5% of female calves born to seropositive dams become latently infected [88]. Accompanied by the lack of periodical testing of newly purchased animals in Egyptian dairy farms and among farmers, and the lack of quarantine measures in these holdings, this can be a major source of introducing infection to dairy farms or retaining the infection.

Serological testing is considered the cornerstone nowadays for brucellosis surveys all over the world [1]. Initial diagnosis is made by the GOVS labs across different Egyptian governorates where suspected cases (positive to screening tests like RBT) are sent to the Brucellosis Research Department, WOAH reference lab, AHRI for further confirmation. The reason for the fluctuation in the Brucellosis prevalence curves, drawn based on the official data obtained from GOVS through published research [40], may result from biased sampling or a low number of tested animals annually. Likewise, the inaccuracy in attributing laboratory results to the initial examination of brucellosis is due to a lack of transparency in the official numbers of examined animals by GOVS labs. Many authors utilize iELISA for serological testing of farms for brucellosis, regarding it as a confirmation test; nevertheless, it functions primarily as a screening test characterized by high sensitivity and low specificity, hence increasing the occurrence of false positives [89]. Also, it is impossible to differentiate between infected and vaccinated animals using available serological tests, and this is another obstacle for disease control. Therefore, Elbauomy et al. [82] and Hegazy et al. [12] confirmed that the success of any control program should consider the efficacy of serological test performance.

Vaccination and vaccine efficacy are other elements of disease control obstacles. Factors contributing to vaccination failure in endemic regions encompass inadequate resources for sustainable vaccination campaigns in low-income areas and the insufficient protection provided by the live attenuated *B. abortus* RB51 vaccine, which is extensively utilized on farms across large ruminant populations despite the predominance of *B. melitensis* circulating among various ruminants in Egypt. This type of rough vaccine gives low protection against *B. melitensis* [92]. *B. abortus* field trains were isolated from miscarried fetuses in cows’ farms vaccinated with RB51 in Egypt [20]. Bacteriological identification and PCR confirmed two *B. abortus* bv1 smooth and two *B. abortus* rough strains. None of the *B. abortus* isolates were identified as RB51 [20].

## 4. Prevalent *Brucella* Strains in Egypt

### Bacteriological Identification and Biotyping

Bacteriological isolation of Brucellae is the gold standard for diagnosis of brucellosis, but this method has several limitations, such as low sensitivity in chronic cases and due to low bacterial load in some samples, it is time-consuming, and with high infection risk for the operator.

Identification of *Brucella* isolates recovered from different samples such as milk, aborted fetuses, lymph nodes, and other tissue specimens of slaughtered animals can be carried out for typing of the causative bacteria with bacteriological and molecular methods [1].

In Egypt, the isolation and identification of *B. abortus* from cattle were reported by many researchers as early as 1943. Then, in 1970 *B. melitensis* was isolated from sheep and goats and *B. abortus* biovar 1 was isolated from cattle and buffaloes. Later on, *B. melitensis* biovar 3, was the most predominant strain that has been isolated from cattle, buffaloes, sheep, goats, and camels [16].

*B. melitensis* cross-infection has been reported in southern Europe [93] and in the Middle East [27,94,95,96]. In contrast, *B. abortus* was isolated from cattle, buffaloes, sheep, and camels in Egypt [14,28,85].

The reappearance of *B. abortus* might be attributed to latent infections or chronic infections that still discharge the organisms intermittently or due to the introduction of infected animals from enzootic countries.

Re-emergence of *B. abortus* as an agent of brucellosis in cattle, the preferred host in Egypt, denotes the lack of continuous surveillance of brucellosis and the need for more accuracy. Continuous monitoring is also very important because of the latent infection with *B. abortus* that might extend for several years. The isolation of the organism from the dairy is of utmost importance in the transmission of the disease to humans, even to those who are considered non-dealers with infected animals.

*B. melitensis* has been mostly distributed in ruminants in the Nile Delta region, including Qalyoubia, Alexandria, Behira, Menofia, Gharbia, Dakahlia, Kafr-Elsheikh, Damietta, Port-Said, Ismaielia, Suez, and Sharkia, and less frequently in Upper Egyptian governorates (Figure 6), such as Giza, Fayoum, Beni-Suef, Minya, Sohag, Quena, Assuit, Luxor, and Aswan. The second most common *Brucella* species isolated from cattle in the Nile Delta governorates and some governorates in Upper Egypt (Giza, Assuit, Beni-Suef, and Aswan) was *B. abortus* biovar 1. Additionally, *B. abortus* biovar 1 was isolated from non-typical host species (sheep) in the governorates of the Nile Delta (Qalyoubia, Alexandria, Menofia, Gharbia, and Suez), as well as in a governorate in Upper Egypt (Aswan). The virulent strain of *B. suis* bv1 was isolated from the milk of a cow (atypical host) in the Menofia governorate, located in the Nile Delta region. Also, it was found in a lymph node of another cow in the Beni-Suef governorate, situated in northern Upper Egypt.

Several authors who wrote about the prevalent *Brucella* species in Egypt reported that the *B. melitensis* biovar 3 is the most prevalent *Brucella* species in bovines, ovines, caprines, and dogs [7,9,10,13,21,22,30,87,97,98,99,100]. *B. abortus* biovar 1 was isolated from cattle [10,14,20,28,95,100,101,102,103] and sheep in some governorates [13,14,85] as shown in Figure 6.

The high incidence of *B. melitensis* biovar 3 infection may be attributed to the fact that in several countries of the Middle East, especially in countries having large mobile flocks of sheep and goats with a high prevalence of brucellosis, the test-and-slaughter policies applicable to individual reactors have proven entirely ineffective and unreliable due to numerous factors. The identified difficulties encountered by the veterinary services are to identify infected animals, vaccinate and follow the infected flocks, and control their movements. On the other hand, the elimination of all positive cases is not practical because of the high cost and the difficult provision of replacement [104]. From the other perspective, the mixed-breed husbandry system practiced in Egypt and most of the Middle East is the main reason for *Brucella melitensis* trans-species transmission from infected sheep and goats to the contact cattle and buffaloes, either in households or the same flock/herd [7,12]. From the epidemiological point of view, *B. melitensis* infection is not associated with storms of abortions among cattle in brucellosis-endemic countries like Egypt.

Another view on the transmission of *B. melitensis* from sheep and goats to cattle and buffaloes can be accepted by reviewing the history of brucellosis during the 1970s and 1980s. During these years, the prevalence of brucellosis in sheep and goats continued to increase with little or nothing performed to control the disease which permitted massive transmission of *B. melitensis* to cattle and buffaloes. The predominance of *B. melitensis* biovar 3 can also be explained by its isolation from rats and dogs obtained from cattle-infected farms [9,25,105].

Swine brucellosis has been reported in Egypt on a serological and bacteriological basis [106]. Interestingly, seven isolates of *B. suis* biovar 1 were recovered from six boars and a sow. Although *B. suis* biovar 1 presence in pigs has been reported previously in Egypt, its current distribution is unknown. Considering the presence of *B. suis* biovar 1 and its reservoir–swine population in various areas of Egypt, more efforts are needed to determine the importance of this animal species as a source of infection to other animal species and humans. *B. suis* bv1 (n = 2), which is highly pathogenic for humans, was isolated from cow milk in the Menofia governorate, Delta region, and the second from a cow in Beni-Suef Governorate, North Upper Egypt [28].

Recently, the DNA of *B. suis* has been identified by real-time PCR in the serum samples of slaughtered pigs [107] and in the serum of slaughtered imported camels in Egypt [80], without biovar identification. Taking into account that Egypt’s governmental surveillance methods exclude camels, dogs, cats, and pigs, neglected animal species from the national brucellosis control surveillance. Hence, these animals, which house dairy farms or cross their fences and come into contact with dairy cattle, can play a significant role in the persistence of infection within a herd and are considered a risk factor for disease transmission to livestock and humans. As a result, the gaps and drawbacks in the control and surveillance programs may be attributed to this fact. The unanticipated biovar 2, a biovar limited to continental Europe, was discovered in slaughtered pigs at El-Basateen abattoir for the first time in Egypt [8,39]. Detection of *B. canis* recently, *B. abortus* and *B. melitensis* in Egyptian native dogs’ blood revealed the role of stray dogs that cross the fences of dairy farms in remerging the disease in Brucellosis-free dairy farms in Egypt [9].

Camels are known to be infected by both *B. melitensis* and *B. abortus* when they are reared in close contact with small ruminants and cattle, respectively [1,108,109].

## 5. Genetic Diversity and Epidemiological Traceability of Local Egyptian *Brucella* Strains Using MLVA-16 and WGS

The MLVA-16 is a method of choice for profiling extremely homogeneous *Brucella* populations and tracking the source of *Brucella* infection in Egypt [110]. It has been effectively employed as a potent tool for discriminating *Brucella* isolates globally [111]. In Egypt, this method has also been widely used to investigate the genetic diversity of local *Brucella* isolates in various studies [7,10,28,85,99].

To go deeper into the epidemiological understanding of the *B. abortus* biovar 1, *B. melitensis* bv3 and vaccine strain RB51 circulating among different hosts in Egypt, MLVA-16 patterns previously published in Egyptian papers were reused to compare them to worldwide genotypes from the public database hosted by the University of Paris-Saclay (https://microbesgenotyping.i2bc.paris-saclay.fr/, accessed on 17 February 2023). Repeat unit (U) numbers were imported into BioNumerics as a character dataset. A clustering analysis was performed using a Minimum Spanning Tree (MST) with categorical distance matrices.

MST has been performed on 62 Egyptian *B. abortus* isolates recovered over nine years (2011–2020) and one strain recovered in 2007. Between 2007 and 2011, no *B. abortus* isolates were recorded (Figure 7). These strains were isolated mostly from cattle as a typical host (n = 48) and less frequently from buffaloes (n = 6), sheep (n = 4), humans (n = 2), one cat (n = 1), and one dog (n = 1) as non-preferential hosts (Figure 7). The biological role of cats and dogs in the transmission of *B. abortus* bv1 has been confirmed by Wareth et al. [38]. Dogs and cats, through direct contact with infected animals and their secretions, contribute to the persistence of *B. abortus* within a herd [38]. *B. abortus* bv1 was isolated from the lymph node of one seropositive sheep and the blood samples of two human cases, one of which had no contact with animals [14]. In the most recent study [85], it was reported that there were three clusters with only one isolate distantly related to the others. One cluster identified a rather widely distributed outbreak strain, which has been repeatedly occurring for at least 16 years with marginal deviations in cgSNP analysis. The other cluster of isolates represented a rather newly introduced strain and the third separate cluster comprising RB51 vaccine related strains, isolated from miscarried material. The highest diversity of alleles was in markers Bruce07 (five alleles) and Bruce04, 16, 18, and 30 (three alleles each), as previously described for *B. melitensis* [85].

The 62 *B. abortus* bv1 strains displayed 13 genotypes, 6 as unique (singleton) and 7 as shared genotypes. The VNTRs and associated metadata of the Egyptian *B. abortus* bv1 strains were compared with MLVA-16 data and metadata of isolates from Africa (n = 3), Asia (n = 236), Central America (n = 1), Europe (n = 148), North America (n = 4), South America (n = 71), unknown origin (N/A) (n = 2), and the *B. abortus* reference strain 544 (Figure 8). Remarkably, most local Egyptian *B. abortus* strains were closely related to those from Europe. Two *B. abortus* strains (Id 46, 15649) isolated from buffalos shared the same genotype (100% identical) with strains recovered from the UK (BCCN#R4C, BCCN#R4D, and BCCN R4#*), Brazil (RS 5) and China (CNBa108) as well as the *B. abortus* reference strain 544 (Figure 8). Most strikingly, *B. abortus* strains DNA 20 and 44 presented the same MLVA genotype as two strains, one from South Korea (KBa101) and one from an unknown location (BCCN#R23). Egyptian *B. abortus* isolates showed a genetic relationship with only one African strain from Zimbabwe (BfR 96), which may be due to the shortage of VNTRs available in the database for African strains used for comparison (n = 3), especially from the neighboring African countries. Egyptian *B. abortus* strains also showed a close genetic association with seven strains from South America (Figure 8).

In previous studies, epidemiological strategies combined with phylogenetic methods provided the high-resolution power needed to demonstrate the persistence of *B. abortus* lineages were introduced more than 100 years ago. The history of cattle introduction as well as dissemination throughout the country may contribute to an understanding of phylogenetic associations in a worldwide and national context. More recent introductions are expected to be more geographically restricted [112]. In this study, the predominant genotype in cattle (Figure 9a,b) was recorded in five different provinces (Alexandria, Delta, Greater Cairo, North Upper and Suez Canal regions), which may correspond to a long lineage persistence. For single genotypes, the importation of animals into Egypt may explain the introduction of particular additional genotypes. Over the last 30 years, Egypt imported cattle for breeding from all parts of the world, including Europe, North and South America and Asia. The list of exporting countries is provided by the Food and Agriculture Organization online statistics (FAOSTAT) (http://www.fao.org/faostat/en/#data/TM, accessed on 17 September 2023). For breeding purposes, Egypt imported cattle from France (1993–2020), Germany (1993–2019), Hungary (1986–2019), Italy (1986–2018), and the Netherlands (1986–2018). Except from these European countries, Egypt imported cattle for slaughter from Croatia, Denmark, Portugal, Spain, and others. Importing cattle for breeding from countries that have been officially brucellosis free last two decades does not rule out the possibility of *Brucella* introduction in the period before the free status was obtained. As several Egyptian *B. melitensis* and *B. abortus* strains have the same genotypes as European *Brucella* strains, it is probable that infected cattle were imported in the 1960s and 1970s. Egypt also imported cattle for breeding from the USA, and Canada (1987–2019). Additionally, Egypt imported cattle for slaughtering from Brazil, Uruguay, and Colombia (South America). From Asia, Egypt imported cattle for slaughtering from South Korea, India, and the Philippines (Asia). Holzer et al. [85] showed that the Egyptian *B. abortus* isolates were most closely related with one strain from Argentina, one from Spain, and the isolate 15649 had the same MLVA genotype as a strain from Portugal, while other shared MLVA profile with strains from the USA. Menshawy et al. [28] found that one of the MLVA-15 patterns of the two *B. abortus* strains used in the study matched existing profiles of strains isolated in the USA, Italy, and Portugal. Wareth et al. [99] stated that two Egyptian *B. abortus* strains isolated from cattle in 2012–2014 had the same genotype as strains isolated from Portugal in 2005 and with high genetic similarity to strains from Italy. Abdel-Hamid et al. [10] reported significant genetic similarities (99%) between *B. abortus* genotypes A1_BenSuef, A2_Sharqia, and A3_Dakhlia with strains isolated from South Korea, Portugal, and Brazil. Based on this information, a probable source of infection is the above-mentioned countries where brucellosis is still endemic, including Asia, Europe, and South America. *Brucella* transmission from cattle could have happened early during the 1960s and 1970s, during the open random cattle importation policy, when most European countries were still endemic for brucellosis. As a result, *Brucella* infection disseminated among ruminants, and a well-established infection occurred among cattle in the years after. Imported fattening cattle for slaughter could be another possible source of *Brucella* transmission from various countries to the Egyptian native breeds as some of them could had been used for breeding during the 1970s, 1980s, and 1990s.

Germany, Greece, France, Italy, and Spain reported the highest number of human cases, ranging from 22 to 35 cases per country. However, this may be explained by the population size of these countries (Germany, 83 million; France, 67 million; Italy, 58 million; Spain, 47 million). A total of 22 Member States and the United Kingdom (Northern Ireland) are currently disease-free for brucellosis in cattle, whereas five zones still perform an eradication program with favorable trends (Bulgaria, Greece, Hungary, Italy, and Portugal).

In South Korea, *B. abortus* is the predominant *Brucella* species in both humans and cattle [113]. Although *B. abortus* has been eradicated from cattle herds in all states of the United States, brucellosis remains endemic in selected non-domesticated bison and elk populations. Transmission of *B. abortus* to domestic cattle herds is rare but has occurred in several cases in the greater Yellowstone Park area that have had contact with infected elk [114]. However, it remains difficult to explain the sources of *B. abortus* transmission to Egyptian livestock, as the animals, except those from the United States (1987–2019), France (1993–2020) and Italy (1986–2018), were all imported for slaughter, but it cannot be ruled out.

Lounes et al. [115] attributed the lineage of *Brucella* strains from Africa and Europe to the long-lasting socio-historical connections of the two continents. Import of live animals is considered a risk factor for *Brucella* genotype diversity [99]. Another scenario that could account for the diversity of Egyptian *B. abortus* strains is the illegal introduction of animals into Egypt via the country’s western and southern borders [12]. Open live animal markets, which allow different animal species to come into contact with each other, across the country, without veterinary inspection and authority, result in the spread of both *B. abortus* and *B. melitensis* through different animal species. The outstanding results of the comparisons of Egyptian *B. abortus* rough strains with the entries from the MLVA Bank for Microbes genotyping database, was the detection of the same MLVA genotype composed of RB51 vaccine-like strains isolated in the Costa Rica, Italy, Portugal and the USA, together with the vaccine reference strain RB51 (Figure 10). These rough *B. abortus* strains (field strains (n = 7); RB51 vaccine strain (n = 1)) that were isolated from the Delta and Alexandria regions (n = 1) during the period from 2012 to 2017 showed 100% genetic similarities with *B. abortus* bv1 strains recovered from Alexandria and North Upper Egypt between 2002 and 2017 (Figure 9a,b). *B. abortus* strains with identical MLVA patterns (same MLVA genotype) isolated from different governorates indicate movement of the pathogen between governorates. This may also reflect the long brucellosis endemicity of Egypt, with an earlier distribution of *B. abortus* strains and high local genetic variation [99]. None of the rough RB51-related *B. abortus* isolates were identified as RB51 vaccine strain by Bruce-ladder PCR performed through the published Egyptian research. The isolation of RB51-related strain was unexpected and would demand further investigations, in order to confirm identification and typing details.

The majority of Egyptian isolates of *B. melitensis* biovar 3 were obtained from non-preferred hosts. These include cattle, buffaloes, camels, and humans. Some *B. melitensis* isolates from cattle and sheep have the same MLVA genotypes (Figure 11). This finding may be attributed to the lack of animal movement control in the country [10,12]. Although sheep and goats are the preferred hosts for *B. melitensis* [1], this *Brucella* species has recently been identified as a possible source of infection for other hosts in Egypt, resulting in cross-species transmission. In the most widespread production systems in the country, where ewes, goats, and cattle are kept together in the same herd or household and cannot be separated during parturition or abortion, the likelihood of *B. melitensis* transmission to non-preferential hosts increases the risk of spread among cattle [10,12]. This is exacerbated by the fact that *B. melitensis* can spread from cattle to cattle in the absence of small ruminants (spill-over infection) [7,12,116].

The cluster analysis of *B. melitensis* biovar 3 from Egypt shows 34 clusters and the reference strain Ether (Figure 12). The central cluster (Figure 12—the pink one) contains the vast majority of Egyptian strains, covering all governorates except the South Upper region. This can be explained by the fact that of the 323 Egyptian strains analyzed, only one is from this region. In the same cluster, we can observe one genotype common to five different areas (Delta region, Greater Cairo, Middle Upper, North Upper, and Suez Canal region), which may correspond to a long persistence of that lineage. The green cluster has two genotypes common to Greater Cairo, Middle Upper, and North Upper governorates. In addition, the blue cluster has one genotype common to the Delta region, Greater Cairo, North Upper, and Suez Canal regions, and two other genotypes common to the Delta region, Greater Cairo, and North Upper regions (Figure 12).

*B. abortus* and *B. melitensis* genotypes included in this review were mainly found in northern Egypt, with almost no strains found in the governorates representing the Egyptian borders and less frequently encountered in Upper Egypt (Figure 9a and Figure 12). This finding may indicate biased sampling and, consequently, a less informative and epidemiologically sound understanding of the true distribution of both *Brucella* species in Egypt.

MST with categorical distance matrices was also performed on *B. melitensis* biovar 3 strains from the public database and compared with previously published 324 *B. melitensis* strains [7,12,28,30,87,99] to determine the genetic relatedness and the possible source of *B. melitensis* outbreak in Egypt (Figure 13). The Egyptian *B. melitensis* strains, as shown in Figure 13, are mostly related to European and African *B. melitensis* strains. Only two Egyptian *B. melitensis* strains originate from the Asian/Europe lineage with one strain from Marocco (pink cluster). These strains, named HEA143 and 126, are shown as letter A in Figure 12. Two Egyptian strains (7 and 154) isolated from cattle form a single lineage (green cluster) are shown as letter B in Figure 12. In addition, one strain is in the same lineage as seven *B. melitensis* bv3 from Portugal (yellow cluster) is shown as letter C in Figure 12. In phylogeny, *B. melitensis* isolates were clustered with African isolates recovered from sheep and humans in Algeria and Tunisia [28]. An Egyptian *B. melitensis* strain recovered from cattle in 2011 was grouped with an Italian strain isolated from a bovine and a human in 2011 and 2013, respectively, and with a Tunisian strain [99]. In a study of 118 *B. melitensis* strains to determine the genetic association of Egyptian *B. melitensis* genotypes with *B. melitensis* VNTRs worldwide, genotypes isolated from the Delta region and Upper Egypt four strains (21-Bm3-Suef, 42-Bm3-Ism, 14-Bm3-Dak, and 59-Bm3-Menof) shared the same MLVA genotype with four genotypes isolated from France earlier in 1975, 1978, 1991 and 1997 (BCCN#91-11–1991, BCCN#78-14–1978, BCCN#75-67–1975, and BCCN#97-11–1997). Similarly, the genotype recovered from Giza Governorate (2-Egy-Bm3-Giza) was identical to an Italian genotype (BCCN#96-72) recovered earlier in 1996 [7]. Sayour et al. [7] listed the genetically related Western Mediterranean genotypes to the local Egyptian genotypes in descending order as France, Italy, Spain, Tunisia, and Algeria.

Based on an in silico MLVA analysis with 16 markers of 136 *B. melitensis* isolates compared with database records, Holzer et al. [87] reported that most local *B. melitensis* strains are closer to strains from Italy and France (Western Mediterranean clade) and only one belongs to American cluster. Similarly, Abdel-Hamid et al. [10] and Hegazy et al. [12] reported high genetic similarity of 53 Egyptian *B. melitensis* strains, mainly with strains from Italy and France.

For breeding purposes, Egypt imported 67 goats (Alpine Chamoisée) and 5451 cattle from France in 1993, and 1119 pregnant heifers in 2020 (FAOSTAT; http://www.fao.org/faostat/en/#data/TM, accessed on 17 July 2024). For the same purpose, Egypt imported pregnant heifers from Italy in 1986 (n = 150), 1999 (n = 6877), 2000 (n = 6942), and 2001 (n = 844). Import of animals for breeding is considered to be the main risk factor for the introduction of new strains, especially from brucellosis-endemic countries. Counting on the better correlation with epidemiological metadata, Whole Genome Sequencing based on core genome SNP analysis provides better discrimination than the MLVA-16 genotyping method for *B. abortus* and *B. melitensis* strains, and, as a result, the source determination and tracking of outbreak strains [85,87]. The Egyptian WGS based on cgSNP *B. abortus* and most of *B. melitensis* genotypes are closely related to the Western Mediterranean clade [85,87]. WGS results based on core-genome SNP analysis affirm the presence of *B. abortus* and *B. melitensis* genotypes with many different outbreak strains, both widely disseminated older strains and newly introduced strains, indicating that the sources of infection are both from movements within Egypt and imports from other countries [85,87].

## 6. Conclusions and Future Directions

In Summary, brucellosis still represents a major constraint to livestock production in Egypt, as it is endemic nationwide in many farm animals as well as in other carriers such as dogs, cats, and rodents, besides its important public health impact. Continuous isolation and identification of *Brucella* isolates on both bacteriological and molecular bases represents the cornerstone for epidemiological evaluation of brucellosis and for tracing back the source of infection. The actual brucellosis status during the years 2000 and 2023 refers to *B. melitensis* biovar 3 and *B. abortus* biovar 1, two prevalent types circulating in different Egyptian governorates. In Egypt, the common husbandry system represented by the predominance of smallholdings that favors the presence of mixed populations of cattle, buffaloes, sheep, and goats in the same yards, the limited success of the official control program, lack of animal registration and identification systems, and absence of official control on animal markets have contributed to the re-emergence of brucellosis with the complicated epidemiological situation.

There is an urgent need for a national survey protocol that would include firstly serological tests for detection of the prevalence in different areas all over the country, in parallel with isolation of the microorganisms from infected animals to genotype circulating Brucellae among different animal species in Egypt. This will be the cornerstone to clarify the complex underlying epidemiologic situation of brucellosis in Egypt. At the same time, cultivation and biotyping of *Brucella* isolates have to be made available for all governorates to be able to send strains to the national bank and to trace back sources of infection for monitoring the efficacy of control protocols.

It is important to mention that eradication of brucellosis is only possible when positive animals are culled, and, additionally, when there is application of biosecurity practices as well as fair compensation policies for owners.

An efficient animal identification and registration system must be applied by the veterinary authorities with complete official control of animal markets in addition to the prohibition of the movement of *Brucella*-positive animals. These are essential steps for the control and eradication of the disease in Egypt. Recognizing area-specific risk factors is essential for effective education campaign planning and developing a comprehensive nation-wide prevention strategy, as risk variables vary by region and country. These campaigns should include academic, veterinary, and community health extension specialists. The measures of the control program must be made mandatory, in addition to a reasonable system of compensation and this has to be implemented by the adequate number of public veterinarians for field work and national serology and bacteriology laboratories all over Egypt. Risk factors and defects of the previously applied protocols must be taken into consideration. Finally, the need for one health approach control protocols with increasing awareness about brucellosis is obvious for the eradication of the disease from Egypt.

The concern in Egypt regarding the implementation of T/S related to S19 vaccination in cattle and Riv 1 vaccination in small ruminants is the selective vaccination. The inadequate coverage of vaccinated animals elevates the risk of infection and undermines the efficacy of the implemented control strategy, which also lacks two fundamental components essential for its success: the restriction of animal movements and proper animal identification. Analysis of MLVA-16 and WGS data demonstrated identical strains of *B. abortus* and *B. melitensis* in cattle and various animal species (trans-species transmission) across different Egyptian governorates, indicating the absence of the fundamental cornerstone (control of animal movement) necessary for the currently applied test-and-slaughter strategy associated with the S19/Riv-1 vaccine control approach.

The most straightforward control option, sometimes the simplest, and the only reasonable control strategy to be applied in high prevalence situations, is mass vaccination of mature animals (both male and female). The comprehensive vaccination of all susceptible animals, regardless of age, may be an effective technique to mitigate brucellosis in cattle, sheep, and goats in Egypt, where the disease is highly endemic (5–10% of flocks/herds are affected). A pragmatic and economical suggestion is to do mass vaccinations of the entire flock or herd using S19 and Rev-1 biennially. The optimal timing is to initiate the administration of both vaccines to decrease their drawbacks before the breeding season, during the latter stages of the calving or lambing season, or lactation. Mass vaccination may be enhanced by implementing proper identifying methods for vaccinated animals to ensure effective follow-up in later years. The entire animal population will be completely immunized after mass vaccination of a moderate duration of 5 to 10 years. The most effective method for evaluating the success of the mass vaccination plan is to show a reduction in human cases in subsequent years. Numerous mass vaccination efforts targeting millions of animals have been implemented or are presently underway in many countries to combat the disease, utilizing this technology, with or without individual tagging. When brucellosis prevalence decreases to minimal levels (less than 1% of infected flocks/herds) through sustained mass vaccination of the population, and if veterinary infrastructure and economic resources are concurrently enhanced, eradication via a test-and-slaughter strategy may become feasible.

Moreover, the testing and slaughter of seropositive animals hinder farmers’ compliance with the national control campaign; consequently, vaccination is more favorable to farmers, specifically if it is effective and affordable, especially when coupled with preferential access to animal markets and less restricted movement rules for vaccinated animals. It is essential to elevate their awareness regarding the significance of sticking to animal movement restrictions mandated by GOVS to enhance farmer compliance with control campaigns. There is a need to increase the farmer’s awareness about the importance of following the control of animal movement restrictions by GOVS through education of the farmers about the risks of brucellosis to both animal and human health, highlighting the economic losses from infected herds and the public health risks. Collaboration with farmers is crucial for the successful implementation of even the most basic control program.

Conversely, verified human cases diagnosed by public health fever clinics in Egypt are reported to GOVS, which tests any animals associated with these cases for brucellosis. Consequently, enhancing public health surveillance for brucellosis could significantly aid in identifying areas of high risk and populations, hence reducing the transmission of infection from these locations.

Governmental authorities must conduct immediate and regular surveillance of small-scale pig herds, stray and housed dogs, and imported and native camels to facilitate the early detection and control of Brucellae dissemination from these neglected animal species to contact animals and humans in Egypt. This can be accomplished in dogs by integrating the private sector into the national surveillance system and routinely collecting samples from private veterinary clinics and shelters for brucellosis testing, following educational efforts to inform pet owners of the significance of this infection. Cluster sampling in animal markets, slaughterhouses, and border quarantine could provide a viable and precise method for sampling camels and pigs for brucellosis diagnosis.

The prevention of brucellosis in dogs primarily involves preventing the introduction of infection into a kennel and disrupting transmission from infected to healthy dogs by the capture, sampling for brucellosis diagnosis, castration, sterilization, and release of infected dogs.

## Figures and Tables

**Figure 1 microorganisms-13-00170-f001:**
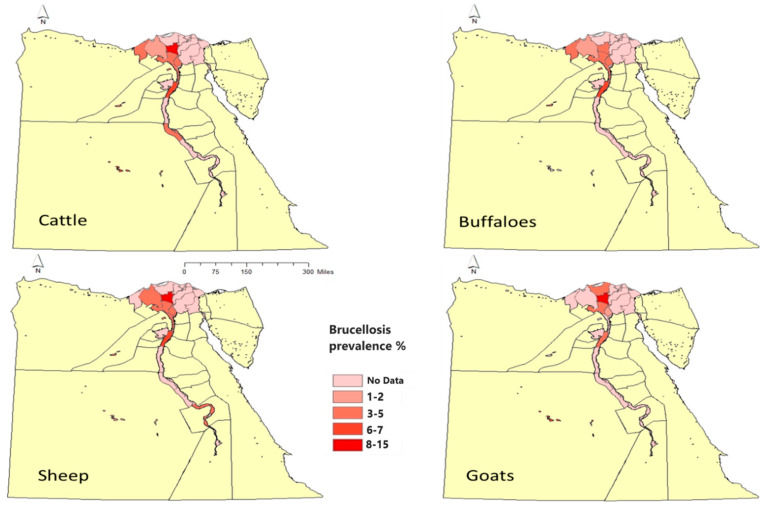
The prevalence estimates of brucellosis among different ruminant species in different governorates of Egypt in the period 2000–2010 were carried out by several authors.

**Figure 2 microorganisms-13-00170-f002:**
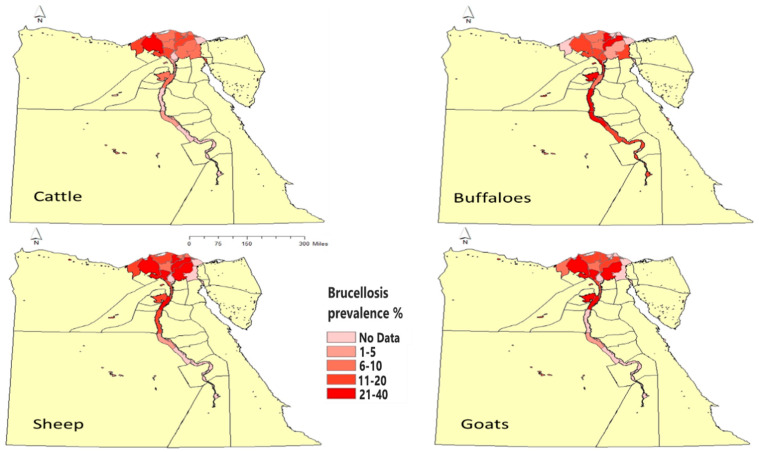
The prevalence estimates of brucellosis among different ruminant species in different governorates of Egypt in the period of 2011–2021 were carried out by various authors.

**Figure 3 microorganisms-13-00170-f003:**
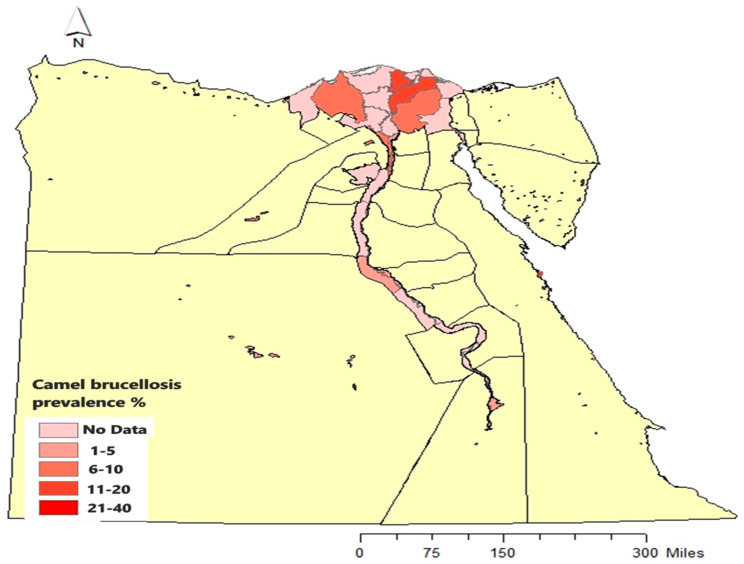
The prevalence estimates of brucellosis among camels in different governorates of Egypt from 2011 to 2023 were carried out by different authors.

**Figure 4 microorganisms-13-00170-f004:**
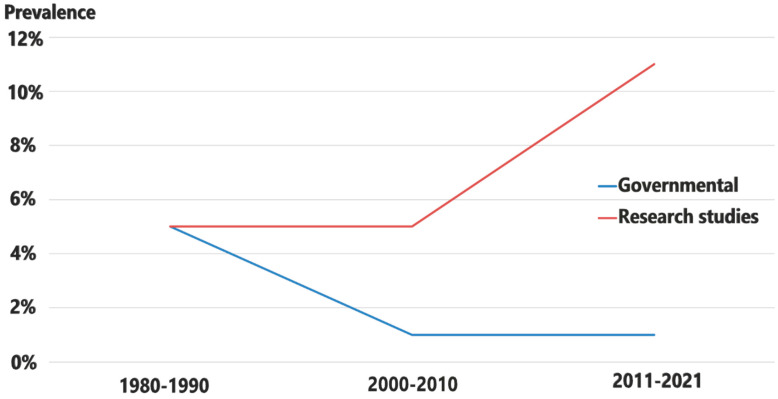
A proposed summary of the actual status of brucellosis prevalence among different animal species in Egypt in the last 30 years.

**Figure 5 microorganisms-13-00170-f005:**
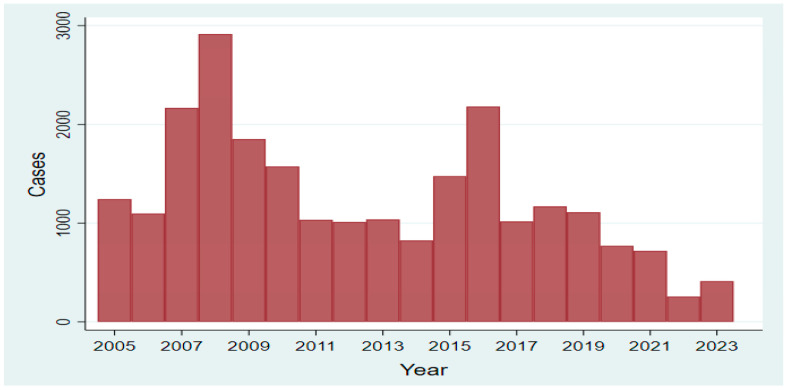
The time series graph represents brucellosis cases among ruminants in Egypt during the period from 2005 to 2023.

**Figure 6 microorganisms-13-00170-f006:**
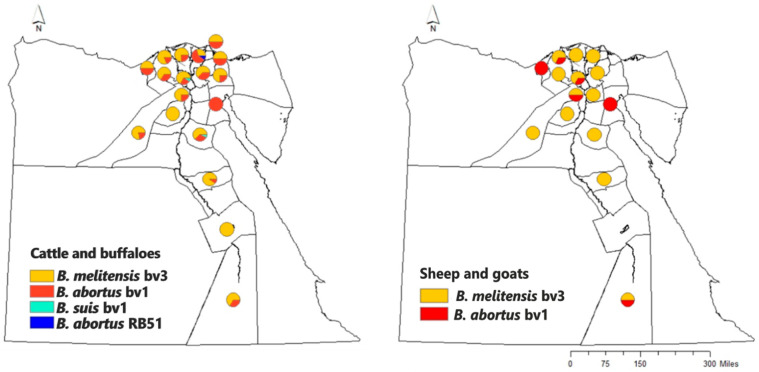
Prevalent *Brucella* species recovered from bovines (cattle and buffaloes) and small ruminants in different Egyptian governorates during 2000–2023.

**Figure 7 microorganisms-13-00170-f007:**
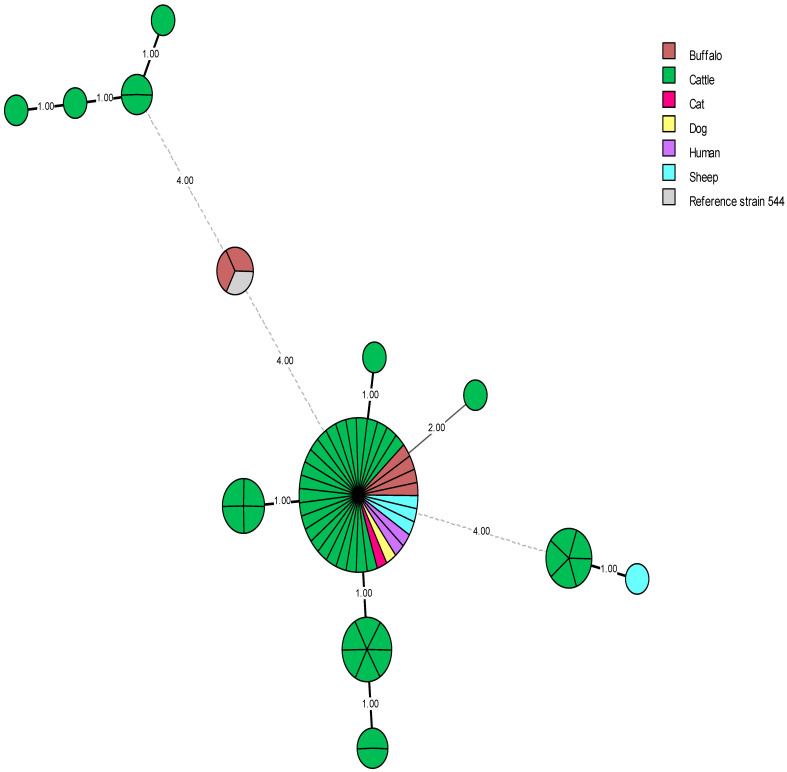
MLVA-16 minimum spanning tree of the Egyptian *B. abortus* biovar 1 strains. Circles represent MLVA-16 genotypes, colored according to host isolation, and the size of the circle indicates the number of strains sharing the same genotype. Egyptian *B. abortus* bv1 strains (n = 62) were analyzed, including the reference strain 544. Each host is represented by a different color: buffaloes (n = 6) in brown, cattle (n = 48) in green, a cat (n = 1) in pink, a dog (n = 1) in yellow, humans (n = 2) in purple, sheep (n = 4) in blue and the reference strain 544 in gray.

**Figure 8 microorganisms-13-00170-f008:**
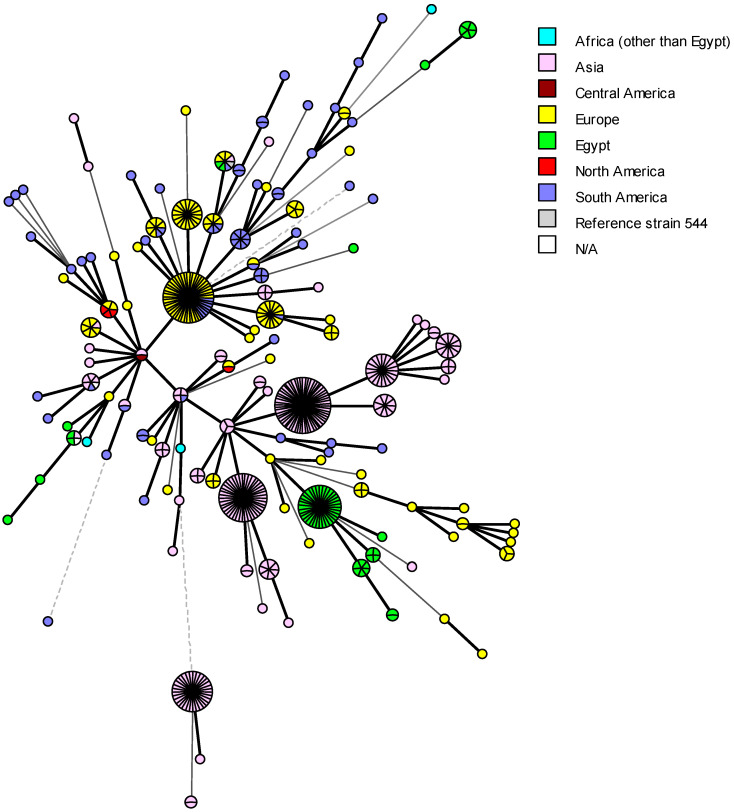
Global distribution of MLVA-16 genotypes of the worldwide *B. abortus* biovar 1 isolates. MLVA-16 minimum spanning tree describing the relationships of 527 *B. abortus* biovar 1 isolates (Africa other than Egypt, n = 3; Asia, n = 236; Central America, n = 1; Europe, n = 148; Egypt, n = 62; North America, n = 4; South America, n = 71; origin not reported sample—N/A, n = 1). Circles represent MLVA-16 genotypes, colored according to the continent of origin, and the size of the circle indicates the number of strains per genotype.

**Figure 9 microorganisms-13-00170-f009:**
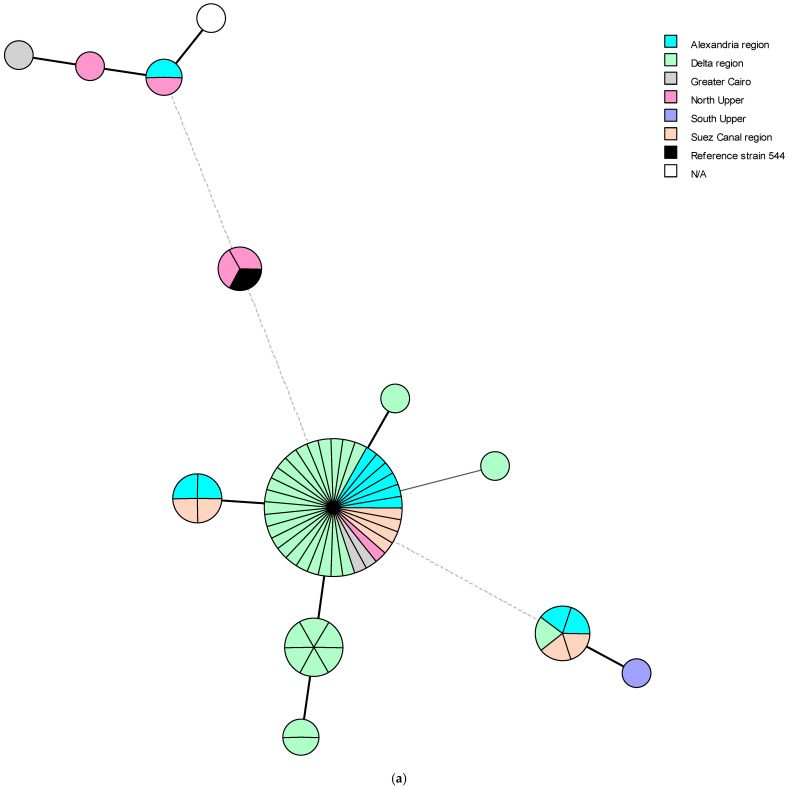
(**a**) MLVA-16 minimum spanning tree of 62 *B. abortus* recovered from different Egyptian governorates. MLVA profiles are colored according to the Egypt region origin: Alexandria region (n = 11), Delta region (n = 33), Greater Cairo (n = 3), North Upper (n = 5), South Upper (n = 1), Suez Canal region (n = 8), N/A (n = 1) and the reference strain 544. (**b**) MLVA-16 minimum spanning tree of *B. abortus* biovar 1 and *B. abortus* rough strains from Egypt. Circles represent MLVA-16 genotypes, colored according to the *Brucella* species, and the size of the circle indicates the number of strains with that genotype.

**Figure 10 microorganisms-13-00170-f010:**
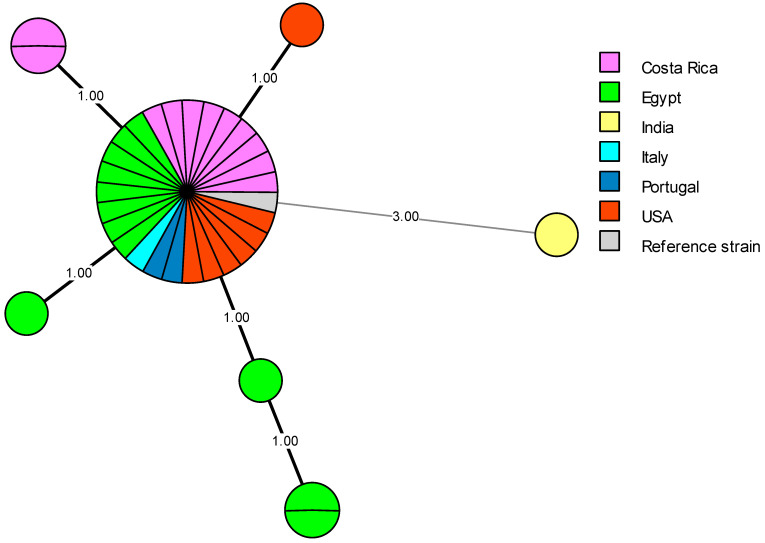
MLVA-16 minimum spanning tree of worldwide *B. abortus* rough strains (Rb51) isolates in Costa Rica (n = 11), Egypt (n = 12), India (n = 1), Italy (n = 1), Portugal (n = 2), and the USA (n = 7). A minimum spanning tree was created using a categorical coefficient with a categorical coefficient. The size of the circles reflects the number of isolates with a particular MLVA genotype. The length of the line reflects the genetic distance between genotypes.

**Figure 11 microorganisms-13-00170-f011:**
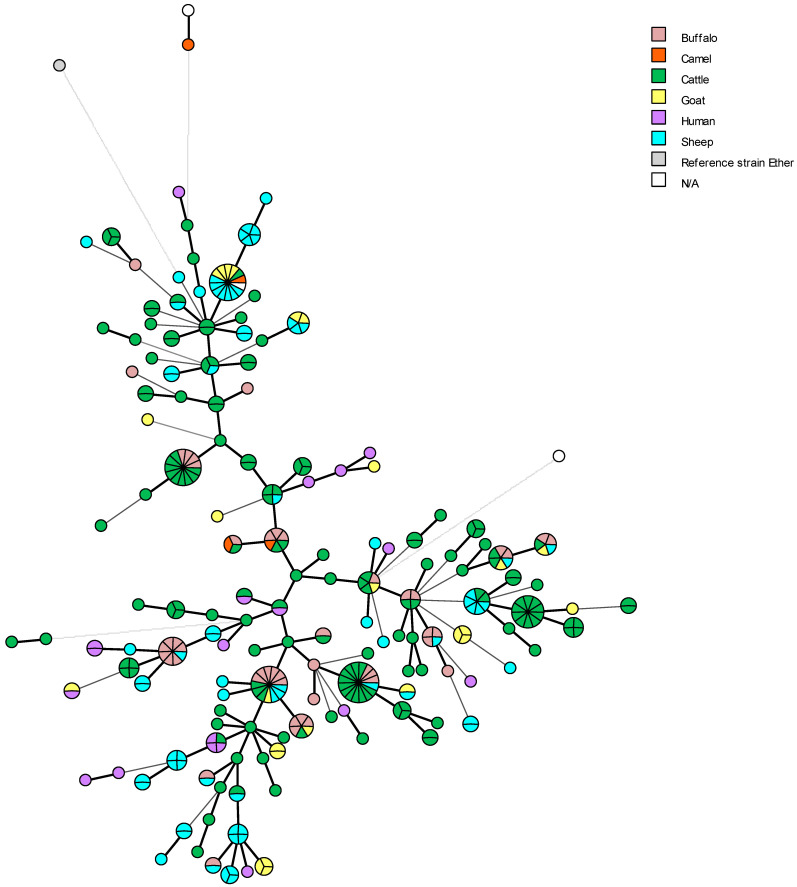
Egyptian distribution of MLVA-16 genotypes of the *B. melitensis* biovar 3 isolates colored by host. A total of 324 Egyptian *B. melitensis* bv 3 strains were analyzed, including the reference strain Ether. Each host is represented by a different color: buffalo (n = 47) in brown, camel (n = 4) in orange, cattle (n = 154) in green, goat (n = 25) in yellow, human (n = 19) in purple, sheep (n = 72) in blue, origin not reported (N/A) sample (n = 3) in white and the reference strain Ether in gray.

**Figure 12 microorganisms-13-00170-f012:**
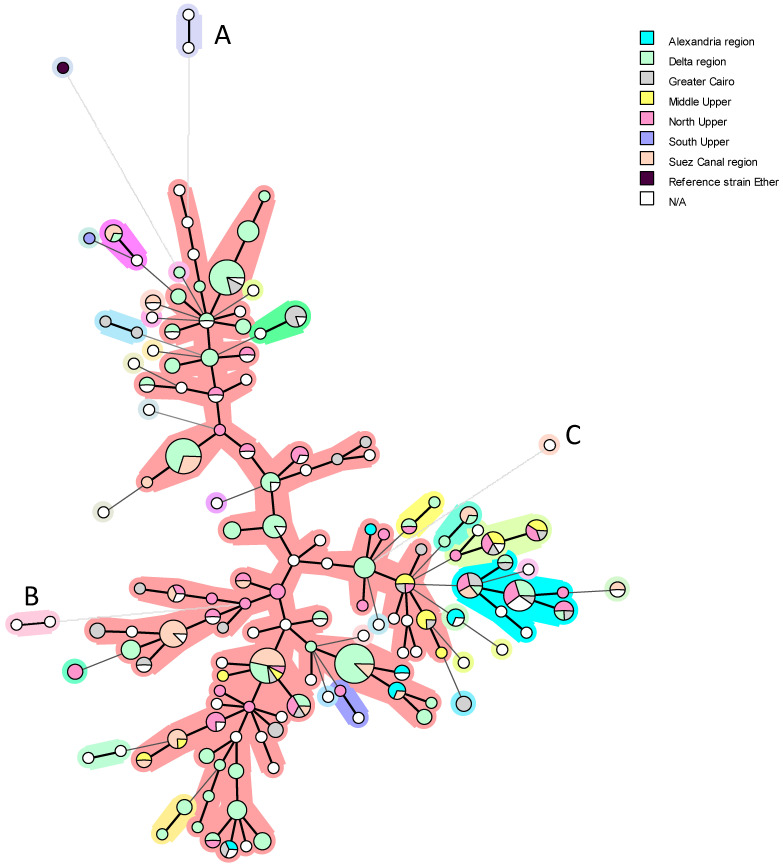
MLVA-16 MST of Egyptian *B. melitensis* biovar 3 was divided into 35 clusters including the reference strain Ether. MLVA profiles are colored according to the Egypt region origin: Alexandria region (n = 7), Delta region (n = 114), Greater Cairo (n = 32), Middle-Upper (n = 14), North Upper (n = 42), South Upper (n = 1), Suez Canal region (n = 36), N/A (n = 78) and the reference strain Ether. For the portioning clustering analysis, nodes closer than 1 have been put into the same partition. Every partition contains at least 1 entry. Every partition contains at least 1 node. Letter (A) corresponds to the strains from the rose lineage in Figure 12; (B) corresponds to the strains from the green lineage in Figure 12; (C) corresponds to the strain in the yellow lineage in Figure 12.

**Figure 13 microorganisms-13-00170-f013:**
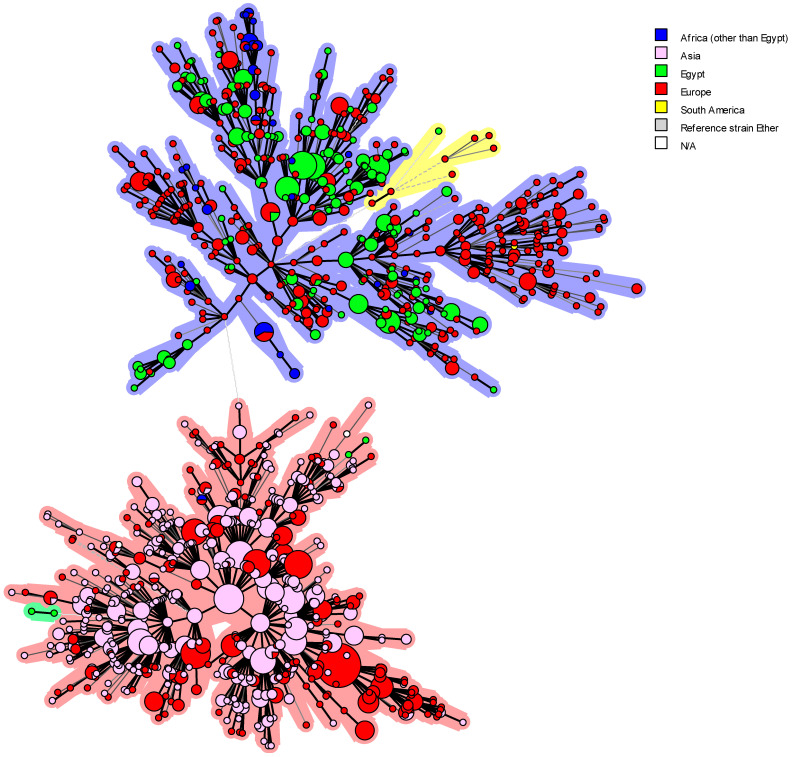
MLVA-16 minimum spanning tree describing the relationships of 2077 worldwide *B. melitensis* biovar 3 isolates. Circles represent MLVA-16 genotypes, colored according to the continent of origin, and the size of the circle indicates the number of strains with that genotype. For the partitioning lineage analysis, nodes closer than 5 have been put into the same partition. Every partition contains at least 1 entry. Every partition contains at least 1 node.

## Data Availability

The original contributions presented in this study are included in the article. Further inquiries can be directed to the corresponding author.

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
