# Peer review of "Animal Brucellosis in Egypt: Review on Evolution, Epidemiological Situation, Prevalent Brucella Strains, Genetic Diversity, and Assessment of Implemented National Control Measures"

_microorganisms, 2025, doi:10.3390/microorganisms13010170_

Round 1
Reviewer 1 Report
Comments and Suggestions for Authors
The article "Animal brucellosis in Egypt: Review on Evolution, Epidemiological Situation, Prevalent Brucella Strains, Genetic Diversity, and Assessment of Implemented National Control Measures" evaluates this disease in various animals, especially ruminants, in Egypt. This manuscript has potential, given the significance of this disease in the country. However, several points need to be addressed in the writing:
References: All references should be cited using square brackets [ ] instead of parentheses ( ). Please modify this throughout the text.
Page 5, Line 229: The phrase "outbreak of Avian Influenza in 2006, swine flu in 2008" refers to events before 2011. Consider rephrasing to "prior events" for clarity.
Page 5, Line 230: The abbreviations FMD, LSD, and BEF should be written in full at their first mention.
Figures 7–13: The names of the animals are too small. Please increase the font size for better readability.
Page 8, Line 361: The term "conclusion" may not be suitable for this section of the manuscript. Consider using a different approach to begin this paragraph.
References: The reference list does not comply with the journal's guidelines. Please review and format all references accordingly.
Author Response
Comment 1:
References: All references should be cited using square brackets [ ] instead of parentheses ( ). Please modify this throughout the text.
Response 1:
Thank you for pointing this out. I agree with this comment. All the reference in the entire text were cited between brackets [ ] and highlighted in red.
Comment 2:
Page 5, Line 229: The phrase "outbreak of Avian Influenza in 2006, swine flu in 2008" refers to events before 2011. Consider rephrasing to "prior events" for clarity.
Response 2:
Thank you for pointing this out. I agree with this comment. The sentence was paraphrased to be " The lack of sufficient governmental information on brucellosis prevalence during this period may be attributed to the prior outbreak events of Avian Influenza in 2006…….etc (page 5, line 209).
Comment 3:
Page 5, Line 230: The abbreviations FMD, LSD, and BEF should be written in full at their first mention.
Response 3:
Thank you for pointing this out. I agree with this comment. The sentences were paraphrased to be "The lack of sufficient governmental information on brucellosis prevalence during this period may be attributed to the prior outbreak events of Avian Influenza in 2006, swine flu in 2008, many foot-and-mouth disease (FMD), lumpy skin disease (LSD), bovine ephemeral fever (BEF) outbreaks in animals, and the human pandemic COVID-19 in 2019 (page 5, second paragraph, lines 210-211).
Comment 4:
Figures 7–13: The names of the animals are too small. Please increase the font size for better readability.
Response 4:
Thank you for pointing this out. I agree with this comment. The font size of all the figures were increased and proofed for better readability. I will upload the new figures with clear and better font size along with the revised review articles.
Comment 5:
Page 8, Line 361: The term "conclusion" may not be suitable for this section of the manuscript. Consider using a different approach to begin this paragraph.
Response 5
Thank you for pointing this out. I agree with this comment. The term conclusion is removed and the sentences were paraphrased accordingly (page 6, second paragraph, lines 271-273).
Comment 6:
References: The reference list does not comply with the journal's guidelines. Please review and format all references accordingly.
Response 6:
Thank you for pointing this out. I agree with this comment. All references were made according to the ACS style guide as per journal request and highlighted in red (pages 18-23, lines 861-1146)
Reviewer 2 Report
Comments and Suggestions for Authors
Introduction:
- While the introduction is detailed, it could benefit from clearer problem framing.
- Start with a broader global perspective on brucellosis and then narrow down to Egypt's specific challenges.
Data Interpretation:
- The disparity between governmental and research estimates of seroprevalence is noted but not analyzed deeply.
- Provide a critical discussion on how discrepancies impact policy-making and control measures.
Discussion:
- The discussion touches on several points but lacks cohesion.
- Conclude each subsection of the discussion with a summary sentence that ties findings to the study's objectives.
Conclusion:
- The conclusion reiterates findings but lacks actionable recommendations.
- Provide clear, actionable recommendations for researchers, policymakers, and veterinary authorities.
Grammar:
- Some sentences are complex, making the text less accessible.
- Simplify sentence structures and ensure clarity. For example, break down sentences with multiple clauses into shorter ones.
Author Response
Comment 1:
Introduction:
- While the introduction is detailed, it could benefit from clearer problem framing.
- Start with a broader global perspective on brucellosis and then narrow down to Egypt's specific challenges.
Response 1:
Thank you for pointing this out. I agree with this comment. Therefore, I added a paragraph to address this point. This paragraph is located in page 2, the second one (lines 53-56).
Comment 2:
Data Interpretation:
- The disparity between governmental and research estimates of seroprevalence is noted but not analyzed deeply.
- Provide a critical discussion on how discrepancies impact policy-making and control measures.
Response 2:
Thank you for pointing this out. I agree with this comment. Therefore, I added a paragraph to address this point. This paragraph is located in page 6, the third one (lines 273-285).
Comment 3:
Discussion:
- The discussion touches on several points but lacks cohesion.
- Conclude each subsection of the discussion with a summary sentence that ties findings to the study's objectives.
Response 3:
Thank you for pointing this out. All summaries belonging to the review sections are addressed in a separate section (Conclusions and Future Directions), pages 15-17, lines 747-845. We added new paragraphs for future directions on suitable control measures and how to apply them based on other reviewers' requests.
Comment 4:
Conclusion:
- The conclusion reiterates findings but lacks actionable recommendations.
- Provide clear, actionable recommendations for researchers, policymakers, and veterinary authorities.
Response 4:
Thank you for pointing this out. However, I disagree with this comment. Future direction and recommendations are addressed in the section of Conclusions and Future Directions), pages 15-17, lines 747-845. However, we added new paragraphs for future directions on suitable control measures (Mass vaccination) and how to apply them based on other reviewers' requests. The new paragraphs are located in lines 797-816, paragraph 1, pointing out why the mass vaccination strategy is the fit strategy under the current epidemiological situation for disease control and the appropriate way to apply such strategy based on successful trials done by many European countries and countries outside Europe using such an approach. Also, we added the second and third paragraphs, lines 828-845, include recommendations for veterinary authorities to involve the neglected animal species not involved in the national surveillance systems by conducting immediate and regular surveillance of small-scale pig herds, stray and housed dogs, and imported and native camels to facilitate the early detection and control of Brucellae dissemination from these neglected animal species to contact animals and humans in Egypt. Besides, we suggested an approach to how the authorities could control the disease in stray and housed dogs. Also, we suggest specific strategies that could be implemented to increase farmer compliance with brucellosis control programs.
Comment 5:
Grammar:
- Some sentences are complex, making the text less accessible.
- Simplify sentence structures and ensure clarity. For example, break down sentences with multiple clauses into shorter ones.
Response 5:
Thank you for pointing this out. I agree with this comment. I revised the review section by section for grammar and type editing using some programs (Grammarly and Quilbot), and we have no problem and encourage any type of editing of the review done by the journal.
Reviewer 3 Report
Comments and Suggestions for Authors
The strengths of the paper lie in its comprehensive review of the epidemiological situation and control measures for animal brucellosis in Egypt, supported by an analysis of genetic diversity using advanced methods like MLVA-16 and whole genome sequencing. Additionally, it provides actionable recommendations for improving brucellosis control through mass vaccination and public awareness, addressing both current gaps and future strategies in combating this endemic zoonotic disease​
To be considered:
The Level of similarity should be reduced by reformulating some phrases
Fig. 4: 1980th -->1980
Fig. 4: Governomental -->Governmental
Fig. 5: What is the difference between "cases" and "cases"?
It could be discussed whether brucellosis is also on the increase in other countries, and why/why not?
Given the long-standing endemic nature of brucellosis in Egypt, what specific evidence supports the feasibility of mass vaccination over test-and-slaughter as a more effective control measure?
How do the authors propose to improve the accuracy and coverage of the national brucellosis surveillance system, especially for neglected species like pigs, camels, and dogs?
What are the potential implications of the identified genetic diversity and trans-species transmission of Brucella melitensis and Brucella abortus on the efficacy of current vaccines and control measures?
What specific strategies could be implemented to increase farmer compliance with brucellosis control programs, particularly in terms of voluntary vaccination and animal movement restrictions?
Given the underreporting and misdiagnosis of human brucellosis, how can public health surveillance be integrated more effectively with veterinary efforts to control the disease in livestock?
The economic impact could be discussed.
Discrepancies between official governmental data and data presented from research studies regarding the prevalence of brucellosis: What is the reason? Please discuss, too.
Author Response
Comment 1:
Fig. 4: 1980th -->1980
Fig. 4: Governomental -->Governmental
Response 1:
Thank you for pointing this out. I agree with this comment. Corrections done for Fig. 4
Comment 2:
Fig. 5: What is the difference between "cases" and "cases"? it's a duplication and this issue is fixed by the statistic program where duplication is removed.
Response 2:
Thank you for pointing this out. I agree with this comment. Both are identical. Brucellosis cases reported to WAHIS-WOAH via passive surveillance are illustrated over time using a combination of line and bar graphs within a single figure. To prevent any misunderstanding, brucellosis cases are represented in bar format, and the figures have been adjusted accordingly by the StataCorp statistical software. I will submit the new Fig. 5 along with the revised review article.
Comment 3:
It could be discussed whether brucellosis is also on the increase in other countries, and why/why not?
Response 3:
Thank you for pointing this out. We appreciate your suggestion. However, it is out of the scope of this review. We adhere to the review objectives, which concentrate on animal brucellosis in Egypt: its evolution, epidemiological status, prevalent Brucella strains, genetic diversity, and the assessment of the national control measures implemented.
Comment 4:
Given the long-standing endemic nature of brucellosis in Egypt, what specific evidence supports the feasibility of mass vaccination over test-and-slaughter as a more effective control measure?
Response 4:
Thank you for pointing this out. I agree with this comment. We addressed the specific evidence that substantiates the feasibility of mass vaccination as a more effective control strategy compared to test-and-slaughter on pages 15 and 16, lines 787-816. Furthermore, addressed on page 7, 1st paragraph, lines 312-318.
Comment 5:
How do the authors propose to improve the accuracy and coverage of the national brucellosis surveillance system, especially for neglected species like pigs, camels, and dogs?
Response 5:
Thank you for pointing this out. I agree with this comment. We discussed enhancing the precision and scope of the national brucellosis surveillance system for neglected species such as pigs, camels, and dogs on pages 16-17, lines 833-845.
Comment 6:
What are the potential implications of the identified genetic diversity and trans-species transmission of Brucella melitensis and Brucella abortus on the efficacy of current vaccines and control measures?
Response 6:
Thank you for pointing this out. I agree with this comment. The underlined paragraph is addressed on page 16, lines 787-796.
The B. abortus S19 vaccination effectively prevented disease transmission and clinical manifestations, including abortion, after natural exposure to B. melitensis (Straten et al., 2016). The cross-protection of a B. abortus S19 vaccination in dairy cattle naturally infected with B. melitensis has been previously proven in multiple studies (Baugés et al., 1991; Straten et al., 2016). The B. abortus S19 vaccine was essential in decreasing herd incidence across several epidemiological and breeding systems before attaining eradication only through Test/Slaughter strategies. Canada, the United States, New Zealand, Australia, and other European Union countries, all possessing substantial cattle populations, systematically employed S19 as a preliminary measure to facilitate eradication using T/S techniques (Blasco et al., 2023). The concern in Egypt regarding the implementation of T/S related to S19 vaccination in cattle and Riv 1 vaccination in small ruminants is the selective vaccination. The RB51 rough vaccine utilized by some private farms provides inadequate protection compared with S19, as field strains of B. abortus and B. melitensis were isolated from aborted vaccinated animals and confirmed through molecular analysis. The inadequate coverage of vaccinated animals elevates the risk of infection and undermines the efficacy of the implemented control strategy, which also lacks two fundamental components essential for its success: the restriction of animal movements and proper animal identification. Analysis of MLVA-16 and WGS data demonstrated identical strains of B. abortus and B. melitensis in cattle and various animal species (trans-species transmission) across different Egyptian governorates, indicating the absence of the fundamental cornerstone (control of animal movement) necessary for the current applied T/S strategy associated with the S19/Riv-1 vaccine control approach.
This paragraph elucidates why mass vaccination is the ideal method for disease control in the current epidemiological context of Egypt, in addition to previously given reasons for the feasibility of mass vaccination over the T/S strategy and its potential implementation.
Comment 7:
What specific strategies could be implemented to increase farmer compliance with brucellosis control programs, particularly in terms of voluntary vaccination and animal movement restrictions?
Response 7:
Thank you for pointing this out. I agree with this comment. We discussed specific strategies could be implemented to increase farmer compliance with brucellosis control programs on page 16, third paragraph, lines 817-827.
Comment 8:
Given the underreporting and misdiagnosis of human brucellosis, how can public health surveillance be integrated more effectively with veterinary efforts to control the disease in livestock?
Response 8:
Thank you for pointing this out. I agree with this comment. The answer of this question is discussed on page 16, 4th paragraph, lines 828-832.
Comment 9:
The economic impact could be discussed.
Response 9:
The economic impact is beyond the purview of this study. It necessitates a separate review, availability of data regarding, and statistical analysis to evaluate updates on the financial costs and the influence of the present control program on the prevalence of brucellosis in ruminants. Such a topic was studied by Eltholth et al., 2017.
Comment 10:
Discrepancies between official governmental data and data presented from research studies regarding the prevalence of brucellosis: What is the reason? Please discuss, too.
Response 10:
Thank you for pointing this out. I agree with this comment. The answer of this question is discussed on page 6, third paragraph, lines 273-285.
Round 2
Reviewer 3 Report
Comments and Suggestions for Authors
The reviewer comments were satisfactorily addressed. The paper can now be recommended for publication.
The publisher should run through the English to make minor language improvements.